# Active sites of copper-complex catalytic materials for electrochemical carbon dioxide reduction

Zhe Weng [1,2,3], Yueshen Wu[2,3], Maoyu Wang[4], Jianbing Jiang[2,3], Ke Yang[2,3], Shengjuan Huo[2,3,5], Xiao-Feng Wang[6], Qing Ma[7], Gary W. Brudvig[2,3], Victor S. Batista[2,3], Yongye Liang[1], Zhenxing Feng[4] & Hailiang Wang [2,3]

Restructuring-induced catalytic activity is an intriguing phenomenon of fundamental importance to rational design of high-performance catalyst materials. We study three copper-complex materials for electrocatalytic carbon dioxide reduction. Among them, the copper(II) phthalocyanine exhibits by far the highest activity for yielding methane with a Faradaic efficiency of 66% and a partial current density of 13 mA cm$^{-2}$ at the potential of −1.06 V versus the reversible hydrogen electrode. Utilizing in-situ and operando X-ray absorption spectroscopy, we find that under the working conditions copper(II) phthalocyanine undergoes reversible structural and oxidation state changes to form ~2 nm metallic copper clusters, which catalyzes the carbon dioxide-to-methane conversion. Density functional calculations rationalize the restructuring behavior and attribute the reversibility to the strong divalent metal ion–ligand coordination in the copper(II) phthalocyanine molecular structure and the small size of the generated copper clusters under the reaction conditions.

[1] Department of Materials Science and Engineering, South University of Science and Technology of China, Shenzhen 518055, China. [2] Department of Chemistry, Yale University, New Haven, CT 06511, USA. [3] Energy Sciences Institute, Yale University, West Haven, CT 06516, USA. [4] School of Chemical, Biological, and Environmental Engineering, Oregon State University, Corvallis, OR 97331, USA. [5] Department of Chemistry, Science Colleges, Shanghai University, Shanghai 200444, China. [6] School of Chemistry and Chemical Engineering, University of South China, Hengyang, Hunan 421001, China. [7] DND-CAT, Synchrotron Research Center, Northwestern University, Evanston, IL 60208, USA. Zhe Weng, Yueshen Wu and Maoyu Wang contributed equally to this work. Correspondence and requests for materials should be addressed to Y.L. (email: liangyy@sustc.edu.cn) or to Z.F. (email: zhenxing.feng@oregonstate.edu) or to H.W. (email: hailiang.wang@yale.edu)

Electrochemical conversion of $CO_2$ using electricity generated from renewable energy sources could provide viable solutions to the development of carbon-neutral fuels. However, $CO_2$ electroreduction is a kinetically slow and diverging reaction that requires a significant magnitude of overpotential and generates a myriad of products[1-3]. Among all the electrocatalyst materials studied thus far for $CO_2$ reduction metal complexes are of distinct importance, because they possess well-defined structures that can be tailored on the molecular level[4-10]. Therefore, there is a significant interest in the development of electrocatalytic materials by deposition of molecules with high catalytic activity and selectivity for electrochemical $CO_2$ reduction[6,11-14]. Cu-based metal organic frameworks (MOFs) have been found to be electrocatalytically active for reducing $CO_2$ to alcohols[14]. Recently, we discovered a Cu porphyrin based heterogeneous electrocatalyst that can reduce $CO_2$ to methane ($CH_4$) and ethylene ($C_2H_4$) in a neutral aqueous electrolyte[15]. Although postmortem analysis reveals that the Cu porphyrin molecular structure remains unchanged after electrolysis, the actual active species responsible for catalyzing $CO_2$ to hydrocarbon conversion has yet to be established.

Many catalyst materials change their structures under reaction conditions[16-18]. The restructuring can be induced by physical conditions such as temperature, pressure, and electrical potential, as well as chemical conditions such as adsorbates and reactants[17-25]. The formed structures with reduced thermodynamic energy under the reaction conditions are responsible for the observed catalytic properties. Metal surfaces are known to alter their atomic arrangements, compositions, and oxidation states under the influences of gas atmosphere and temperature[21]. Numerous studies have been reported on molecular complexes of earth-abundant metals as pre-catalysts for water oxidation[26]. Recently, transition metal sulfides, selenides, and phosphides have been found to reconstruct themselves to the corresponding oxides or (oxy)hydroxides and effectively catalyze electrochemical oxygen evolution[27,28]. Examining the restructuring of catalyst materials is crucial to understanding structure-reactivity correlations and to designing better catalysts.

In situ and operando characterization techniques are highly useful in uncovering catalyst restructuring phenomena, as they can provide chemical and physical information under reaction conditions[29-33]. Under certain circumstances where the reconstructed catalysts are subject to further structural changes when the reaction conditions are removed, operando characterization is necessary to identify the real catalytically active species. In this regard, in-situ and operando X-ray absorption spectroscopy (XAS) is particularly powerful, as the X-ray absorption near edge structure (XANES) can reveal the oxidation state of the element of interest, and extended X-ray absorption fine structure (EXAFS) is capable of probing the influence from the local coordination environment. For example, sub-monolayer $VO_x$ anchored on an $\alpha$-$Fe_2O_3$ powder surface undergoes redox-induced dynamic changes of atomic structure and oxidation state, as revealed step-by-step with in-situ XAS[34]. In another study, XAS is used to determine the cation occupation of the octahedral and tetrahedral sites in spinel oxides[35], which is then identified as a property descriptor of the catalytic activity of these materials for the oxygen reduction and evolution reactions[36].

Here we report the restructuring of three molecularly structured Cu catalysts under electrochemical $CO_2$ reduction conditions as probed by in-situ and operando XAS, and the correlation of the catalyst structures to the observed catalytic properties. The three Cu-complex materials, namely copper(II) phthalocyanine (CuPc), copper(II) benzene-1,3,5-tricarboxylate (btc) MOF (HKUST-1), and copper(II) 1,4,8,11-tetraazacyclotetradecane chloride ([Cu(cyclam)]$Cl_2$), all show catalytic activity toward $CO_2$

reduction to $CH_4$, working as heterogeneous catalysts in 0.5 M $KHCO_3$ aqueous electrolyte. Among them, the CuPc catalyst exhibits the highest activity and selectivity; the partial current density and Faradaic efficiency of the $CH_4$ product reach 13 mA cm$^{-2}$ and 66% at $-1.06$ V vs the reversible hydrogen electrode (RHE), respectively. Thus, CuPc represents one of the most efficient catalysts for electrochemical reduction of $CO_2$ to $CH_4$. In situ and operando XANES and EXAFS studies reveal that the CuPc molecules restructure to metallic Cu clusters with a size of ~2 nm under the working conditions and the Cu nanoclusters convert back to the original CuPc structure upon release of the negative electrode potential. In contrast, HKUST-1 and [Cu(cyclam)]$Cl_2$ irreversibly decompose to form much larger Cu nanostructures. These comparisons indicate that the good performance of the CuPc catalyst originates from the reversible formation of Cu nanoclusters. Further analysis provides deeper understanding toward designing metal-complex molecular structures for controllably generating active species under reaction conditions to catalyze desirable chemistry.

## Results

**Electrocatalytic measurements**. CuPc is a molecular complex with the $Cu^{2+}$ ion coordinated by the conjugated planar $Pc^{2-}$ ligand (Fig. 1a). HKUST-1 is a MOF with Cu(II) nodes coordinated by negatively charged btc linkers (Fig. 1b). [Cu(cyclam)]$Cl_2$ features a $Cu^{2+}$ ion coordinated by a non-conjugated charge-neutral ligand (Fig. 1c). The three materials were each mixed with mildly oxidized multi-wall carbon nanotubes (CNTs)[20,37,38] to form a catalyst layer on electrodes for electrocatalytic measurements in $CO_2$-saturated 0.5 M $KHCO_3$ aqueous solution. Controlled-potential electrolysis was performed with the working electrode potential being varied in the range between $-0.76$ and $-1.36$ V vs RHE at 0.1 V intervals. At relatively lower overpotentials, the major $CO_2$ reduction products over the CuPc catalyst are formic acid (HCOOH), $C_2H_4$, and CO (Fig. 1d), with the Faradaic efficiencies being 25, 13, and 6% at $-0.86$ V, respectively. As the electrode potential goes to $-0.96$ V, $CH_4$ becomes the dominant $CO_2$ reduction product. At $-1.06$ V, a maximum Faradaic efficiency of 66% together with a partial current density of 13 mA cm$^{-2}$ is achieved for $CO_2$ conversion to $CH_4$ (Fig. 1d, e), corresponding to a $CH_4$ formation rate of 0.36 mmol s$^{-1}$ g$_{CuPc}$$^{-1}$ and 0.86 μmol C$^{-1}$. The HKUST-1 and [Cu(cyclam)]$Cl_2$ catalysts are also active for catalyzing $CO_2$ electroreduction to $CH_4$. However, the onset potentials are 100 ~ 200 mV more negative than that of the CuPc catalyst. HKUST-1 reaches a maximum Faradaic efficiency of 27% at $-1.16$ V with a partial current density of 4.4 mA cm$^{-2}$ (Supplementary Fig. 1), whereas [Cu(cyclam)]$Cl_2$ electrodes does so at $-1.26$ V with the Faradaic efficiency and partial current density being 15% and 2.8 mA cm$^{-2}$ (Supplementary Fig. 2). The Faradaic efficiencies and partial current densities of the gas-phase products over the three catalyst electrodes at $-1.06$ V are compared in Fig. 1f, g. It can be clearly discerned that CuPc is a much more active and selective electrocatalyst than HKUST-1 and [Cu(cyclam)]$Cl_2$ for $CO_2$ reduction to $CH_4$.

**In situ and operando XANES measurements**. To probe the structural and oxidation state changes of these Cu-complex electrocatalysts as they perform $CO_2$ reduction, we carried out in-situ XAS measurements (Supplementary Fig. 3) under the same electrochemical conditions. During the measurements, the working electrode potential was first deceased in steps from the open circuit voltage (OCV, ~0.8 V vs RHE) to $-1.06$ V vs RHE, and then increased back to 0.64 V. Each potential was held for at least 1 h until the XAS spectra were recorded. Thus, the results

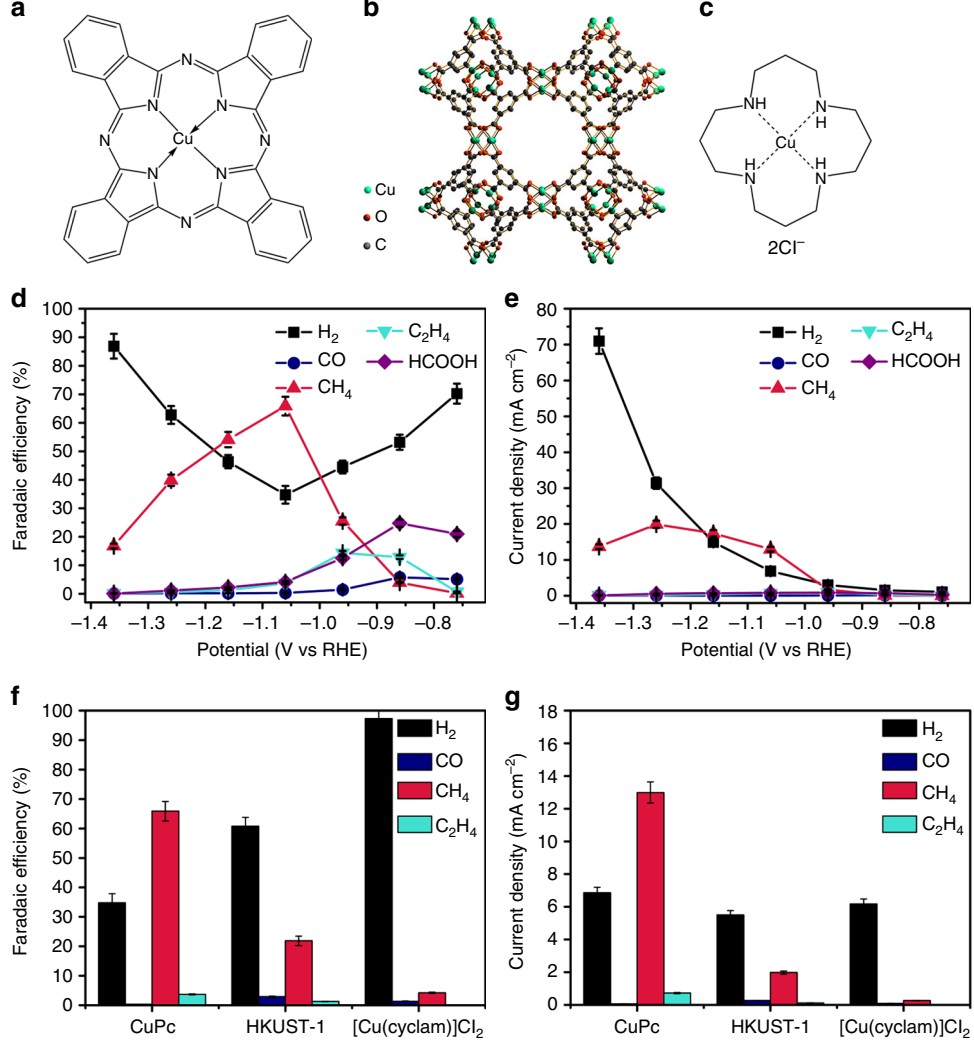

**Fig. 1** Molecular structures of three Cu-complex materials and their electrocatalytic performance for CO₂ reduction. Molecular structures of **a** CuPc, **b** HKUST-1, and **c** [Cu(cyclam)]Cl₂. Potential-dependent **d** Faradaic efficiencies and **e** partial current densities of products for CO₂ electroreduction reaction catalyzed by CuPc. Comparison of **f** Faradaic efficiency and **g** partial current density distributions among CO₂ electroreduction reactions catalyzed by the three materials at –1.06 V vs RHE. Error bars represent the SD from multiple measurements

reflect stable states under the electrochemical conditions. Under the initial OCV conditions, all the three catalysts show a characteristic Cu(II) peak (1 s → 3d transition) at ~ 8,985 eV in the corresponding normalized Cu K-edge XANES spectrum (Fig. 2a, d,g). As the potential applied to the CuPc electrode is decreased to −0.66 V, a small absorption peak appears at ~ 8,981 eV in the spectrum (Fig. 2a), indicating the formation of Cu(I). Another peak at ~ 8,980 eV starts to develop at −0.86 V, which corresponds to Cu(0). At −1.06 V, where the highest Faradaic efficiency for CO₂ reduction to CH₄ is reached, the XANES is dominated by the Cu(0) feature. The spectral evolution and absorption peaks can be discerned more clearly in the derivative curves of the XANES spectra (Fig. 2b). It is worth noting that the Cu(II) peak persists throughout all the applied potentials (Fig. 2a, b), which means that not all the Cu(II) centers are converted to lower oxidation states under the CO₂ reduction conditions. Upon switching the electrode potential back to 0.64 V, the Cu(0) peak disappears and the XANES spectrum is almost restored to that under the initial OCV conditions (Fig. 2a, b), suggesting that the potential-induced oxidation state changes for the CuPc catalyst are reversible. In contrast, neither the HKUST-1 nor the [Cu(cyclam)]Cl₂ exhibits such a reversibility, though both of them are converted to Cu(0) at −1.06 V (Fig. 2d,e,g,sh). The cycled

HKUST-1 electrode mainly contains Cu(I) species while the [Cu(cyclam)]Cl₂ electrode is dominated by metallic Cu (Fig. 2d,e,g,h and Supplementary Fig. 4).

**In situ and operando EXAFS measurements.** To examine the local coordination environment changes, we performed in-situ EXAFS measurements. At the working potential of −1.06 V, all the three catalysts exhibit a characteristic metallic Cu–Cu bond peak at $R = \sim 2.2$ Å in the corresponding Fourier-transformed EXAFS spectrum (Fig. 2c, f, i), which is consistent with the appearance of Cu(0) observed in the XANES spectra. In particular, the EXAFS spectra of HKUST-1 and [Cu(cyclam)]Cl₂ at −1.06 V are similar to that of the Cu metal standard in the entire $R$ and $k$ ranges (Fig. 2f, i and Supplementary Figs 4, 5), suggesting formation of bulk Cu metal. As the electrode potential returns to 0.64 V, the metallic Cu originated from HKUST-1 is oxidized to Cu₂O, which is supported by the similarity to the EXAFS spectrum of the Cu₂O standard (Fig. 2f and Supplementary Fig. 5). The Cu(0) species derived from [Cu(cyclam)]Cl₂ form a reddish metallic sheen on the electrode (Supplementary Fig. 6), which results in reduced and noisy signals in the XAS spectrum recorded at −1.06 V (Supplementary Fig. 4). In contrast, the CuPc

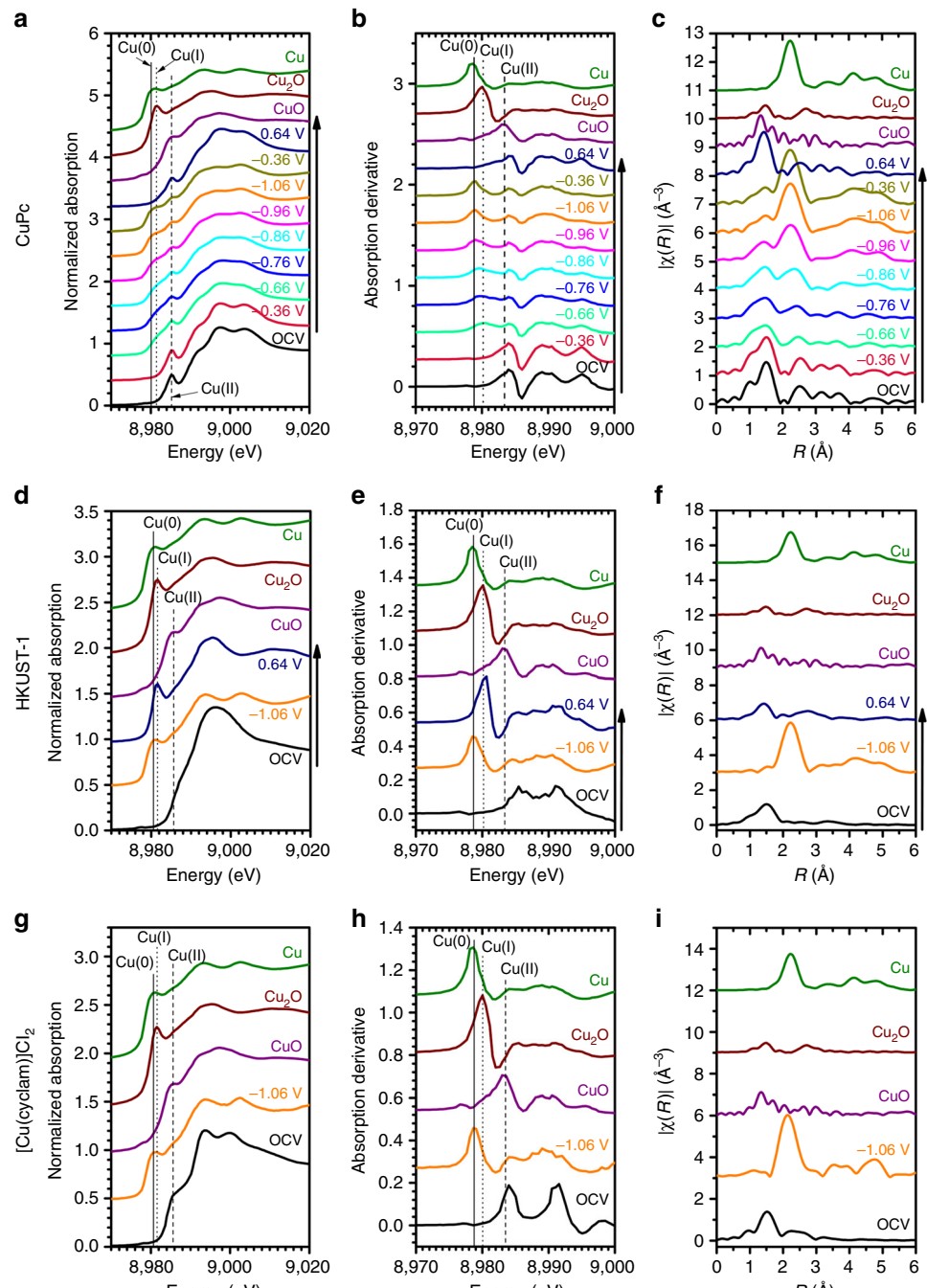

**Fig. 2** In situ XAS measurements under electrocatalytic reaction conditions. **a** Cu K-edge XANES spectra, **b** first-order derivatives of the XANES spectra, and **c** Fourier-transformed Cu K-edge EXAFS spectra for CuPc. Similarly, the corresponding XAS spectra for HKUST-1 are plotted in **d**–**f** and for [Cu(cyclam)]Cl$_2$ in **g**–**i**. The slight off-alignment of the Cu(II) peaks in the XANES derivatives **b**, **e**, **h** with that of CuO is mainly due to the different local geometries of the Cu complexes and CuO

catalyst shows good structural reversibility upon potential cycling. When the electrode potential is negatively polarized toward −1.06 V, the amplitude of the Cu–Cu peak in the EXAFS spectrum increases, indicating gradual formation of metallic Cu species (Fig. 2c). At −1.06 V, in addition to the Cu–Cu peak, which has reached its maximum amplitude, there remain peak features in the $R = 1 \sim 2$ Å range, which can be assigned to the CuPc structure. This suggests that a possible combination of metallic Cu and CuPc molecules exists at the working potential and is consistent with the coexistence of Cu(0) and Cu(II) observed in the XANES spectra. The EXAFS spectrum of the

CuPc catalyst at the final 0.64 V highly resembles that at the initial OCV (Fig. 2c).

**Ex situ XRD and SEM characterizations**. The structural changes are also reflected in ex-situ X-ray diffraction (XRD) and scanning electron microscopy (SEM) characterizations. After the in-situ XAS measurements, the HKUST-1 and [Cu(cyclam)]Cl$_2$ catalyst electrodes have lost the diffraction patterns of the original Cu-complex materials, but exhibit diffraction peaks of Cu$_2$O and Cu, respectively (Fig. 3a, b). Concomitantly, SEM imaging reveals

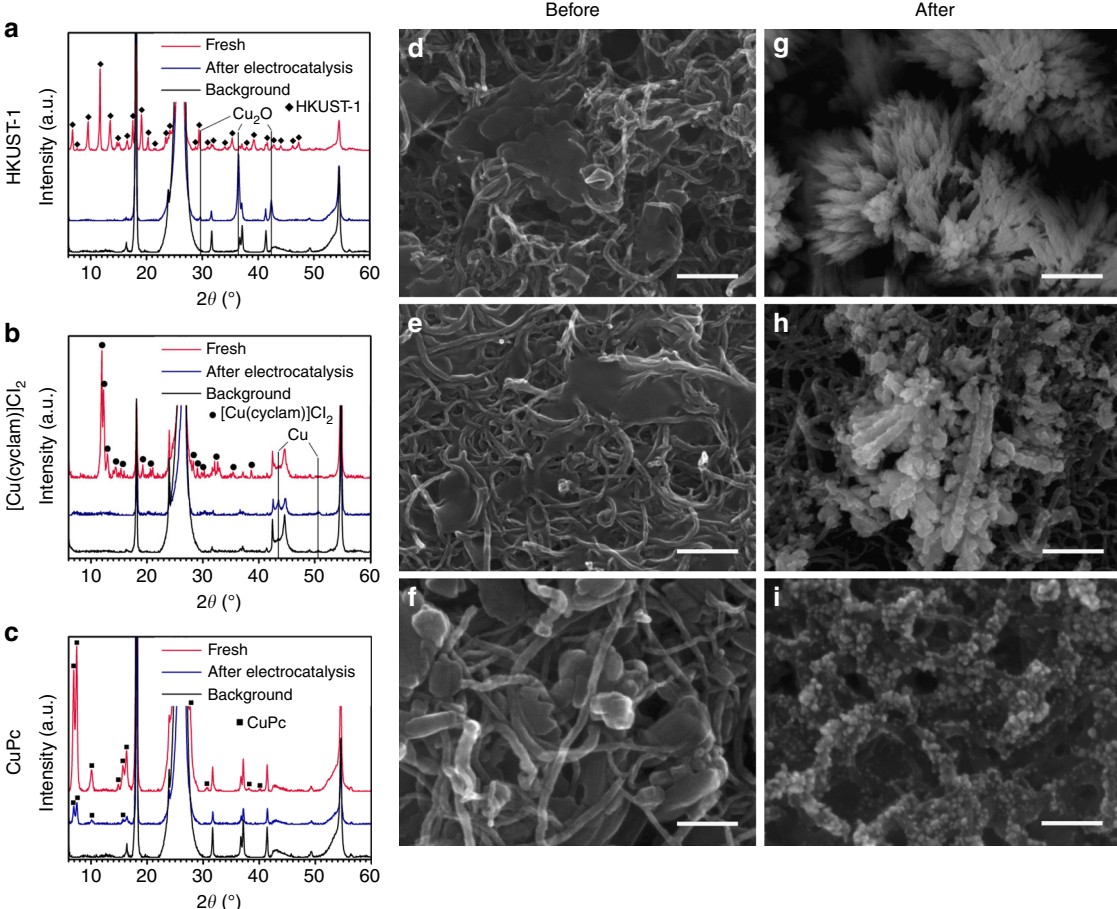

**Fig. 3** XRD and SEM characterizations of the three catalyst materials before and after electrolysis. XRD patterns of **a** HKUST-1, **b** [Cu(cyclam)]Cl₂, and **c** CuPc before and after electrocatalysis. The background diffraction patterns are from carbon paper substrates. SEM images of the **d**, **g** HKUST-1, **e**, **h** [Cu(cyclam)]Cl₂, and **f**, **i** CuPc catalyst materials **d–f** before and **g–i** after electrocatalysis. Scale bars: **d**, **e**, **g**, **h** 200 nm and **f**, **i** 100 nm

morphological changes from the original submicron-sized particles (Fig. 3d, e) to the final dendritic nanostructures (Fig. 3g, h). Unlike HKUST-1 and [Cu(cyclam)]Cl₂, the CuPc catalyst electrode shows no existence of Cu₂O or Cu (Fig. 3c), agreeing with the in-situ XAS results that the CuPc structure is recovered after the working electrode potential is returned to 0.64 V. The remaining CuPc diffraction peaks are likely due to the remaining CuPc crystals that have not experienced restructuring. It is interesting to note that the cycled CuPc electrode features a microstructure of ~ 10 nm-sized nanoparticles well dispersed on the surface of CNTs (Fig. 3i), obviously different from that of the original CuPc (Fig. 3f). The observed morphological changes are a result of the restructuring processes taking place during the potential cycle even though the original CuPc molecular structure is recovered after the cycle.

**EXAFS modeling and analysis**. To gain further insights into the restructuring of the CuPc catalyst under the electrochemical CO₂ reduction reaction conditions, we performed model-based analysis to quantify the in-situ EXAFS results. The fitted spectra are shown in Fig. 4a, b and Supplementary Fig. 7, and the fitting parameters related to the major scattering paths (Supplementary Fig. 8) are listed in Supplementary Table 1. Independent parameters of coordination number (CN) are assigned to every scattering path for each spectrum separately. As shown in Supplementary Table 1, the Cu–N and Cu–C (belonging to CuPc) CNs decrease quickly while the Cu–Cu (belonging to metallic Cu) CNs gradually increase with the decrease of the applied potential.

Negligible changes in the scattering path lengths are found. No Cu metal components can be fitted into the EXAFS spectra recorded at potentials of −0.36 V or higher. At −0.66 and −0.76 V, small Cu–Cu CNs are obtained. Much larger Cu–Cu CNs are obtained as the potential is switched to −0.86 V or lower. As the potential is switched back to 0.64 V, the CuPc component starts to dominate the spectrum again with the CNs recovered to the values obtained under the initial OCV conditions. The potential-dependent first-shell Cu–Cu CNs are plotted in Fig. 4c. In combination with the above XANES analysis, the EXAFS fitting results can be rationalized as follows. The CuPc structure is predominant at OCV and −0.36 V. At −0.66 and −0.76 V, the CuPc structure starts to change with the Cu(II) partially reduced to Cu(I) and thus the CuPc CNs decrease but almost no Cu(0) component is observed. At −0.86 V or lower potentials, Cu(II) and Cu(I) are converted to Cu(0) and the metallic Cu phase nucleates and grows as evidenced by the increasing Cu–Cu CNs. The overall consistency between the XANES and EXAFS analysis demonstrates the validity of our results on the restructuring of the CuPc electrocatalyst under the CO₂ reduction conditions.

We further determine the size of the metallic Cu species reversibly generated by CuPc under the electrochemical conditions. As CNs are sensitive to particle size in the nanometer regime[39,40], it is possible to estimate the size of the formed metallic Cu species based on the Cu–Cu CNs. Following the strategy in the previous reports[39,40], we built a cuboctahedral model (typically adopted for face-centered cubic metal nanoparticles)[41] to obtain the size-dependent Cu-Cu CNs (Supplementary

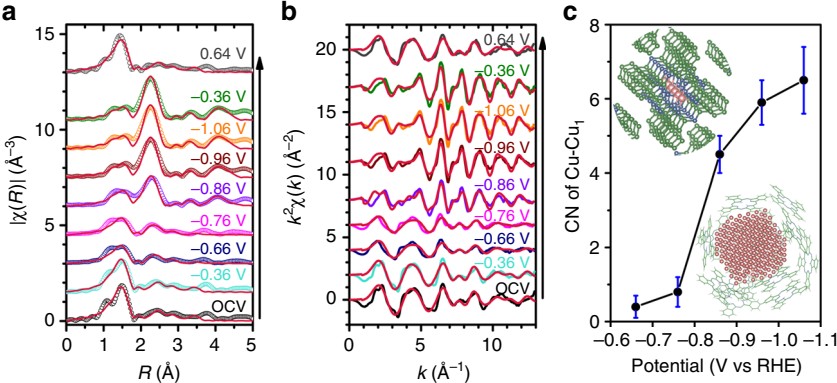

**Fig. 4** Fitting results of the EXAFS spectra of the CuPc catalyst at different potentials in $CO_2$-saturated 0.5 M aqueous $KHCO_3$. Fitted **a** R-space and **b** k-space EXAFS spectra (red traces) of the CuPc catalyst. The experimental data are also plotted for comparison. **c** First-shell Cu–Cu CNs of the CuPc catalyst at different potentials. The upper left inset shows the CuPc crystal structure, and the lower right inset illustrates a possible configuration of the Cu nanoclusters generated under the electrocatalytic conditions. Color key: green—C; blue—N; pink—Cu. Error bars represent the uncertainty of CN determination from the EXAFS analysis

Fig. 9, Supplementary Table 2). As metallic Cu and CuPc coexist on the electrode, the compositional change would also affect the nominal Cu–Cu CNs derived from the XAS analysis[42]. We thus performed linear combination fitting for the XAS spectra at −1.06 V using the reference spectra of Cu foil and CuPc powder. (Supplementary Fig. 10). It shows that there are approximately 20% of CuPc and 80% of metallic Cu in atomic ratio at −1.06 V. The measured CNs were then corrected using this ratio for the compositional effect. By comparing the CNs of the cuboctahedral model nanoparticles with the corrected CNs, we are able to estimate the size of the metallic Cu species formed at −1.06 V vs RHE to be 2 ± 1 nm. Taken together, our results depict a clear picture for the restructuring-induced electrocatalytic activity of CuPc. At −0.86 V, Cu nanoclusters start to form from CuPc demetallation. At the optimum working potential of −1.06 V where the highest $CO_2$-to-$CH_4$ Faradaic efficiency is achieved, the Cu nanoclusters reach an average size of ~2 nm. The corresponding $CH_4$ production rate and turnover frequency (TOF) are 3.2 mmol $s^{-1}$ $g_{Cu}^{-1}$ and 0.39 molecules $s^{-1}$ $site^{-1}$, respectively. Given the known properties of metallic Cu for electrochemically reducing $CO_2$ to hydrocarbons, it is reasonable to believe that the Cu nanoclusters generated from CuPc under the working conditions are most likely the active species for the catalysis. The small size of the Cu nanoclusters generated in-situ appears to be a major contributor to the high current density and selectivity of the CuPc catalyst for electrochemical $CO_2$ conversion to $CH_4$. Although the size dependence of electrochemical $CO_2$ reduction catalyzed by Cu nanoparticles is still under some debate, more studies appear to support the conclusion that smaller particle sizes and more low-coordination surface sites favor $CO_2$ reduction to $CH_4$[43–45]. Shape could be another structural factor responsible for the observed catalytic properties[44,46–50], although at the current stage analyzing the shape of the 2 nm-sized Cu clusters existing under electrochemical conditions is beyond our capability.

## Discussion

CuPc exhibits a different restructuring behavior and thus different catalytic properties from HKUST-1 and [Cu(cyclam)]$Cl_2$. CuPc reversibly forms Cu nanoclusters under the reaction conditions, whereas the latter two irreversibly decompose to form dendritic Cu nanostructures with much larger sizes. Consequently, the CuPc catalyst shows a lower overpotential, higher selectivity, and larger current density for electrochemical $CO_2$

conversion to $CH_4$. We also find that our previously reported Cu porphyrin catalyst operates following a similar reversible restructuring scheme as the CuPc (Supplementary Fig. 11)[15]. To understand more about the distinct restructuring behavior of CuPc, we performed density functional theory (DFT) calculations on the thermodynamics of the reductive demetallation and recovery of the molecular CuPc structure. Plausible thermodynamic pathways (Supplementary Note 1, 2) are constructed for the two processes. The calculation results reveal that the standard reduction potential of CuPc demetallation is 0.23 ~ 0.35 and 0.53 V more negative than those for [Cu(cyclam)]$^{2+}$ and HKUST-1, respectively (Supplementary Table 3), pointing to the higher thermodynamic stability of the CuPc structure than the other Cu complexes. The reversible Cu nanocluster formation in the CuPc case is rationalized on the basis of the intrinsic instability of the small nanoclusters. The critical diameter of a Cu nanocluster, below which the reverse reaction of the CuPc demetallation process can be spontaneous under OCV conditions, is calculated to be 14 nm. These thermodynamic calculations suggest that the metal ion-ligand binding affinity of a Cu complex influences the threshold potential as well as the reversibility of the reductive demetallation process. Although these results are qualitatively consistent with our experimental observations, we note that the restructuring process may involve other important factors such as the solubility of the demetallated ligand and the electronic structure of the complex. With the speculation that the demetallated phthalocyanine ligands must be in the vicinity of the Cu nanoclusters, we sketched a schematic model (Fig. 4c lower right inset) to qualitatively illustrate a possible spatial configuration of the active species derived from CuPc (Fig. 4c upper left inset) under the working conditions. The presence of the ligands may be an important contributor to the observed reversible restructuring behavior and high catalytic activity for $CO_2$ conversion to $CH_4$[51,52].

Restructuring of Cu complexes in electrocatalytic materials for $CO_2$ reduction to $CH_4$ has been elucidated by in-situ and operando XAS characterization of representative Cu complex structures (CuPc, HKUST-1 and [Cu(cyclam)]$Cl_2$) probed under electrochemical reaction conditions. The highest activity and selectivity of CuPc for catalyzing $CO_2$-to-$CH_4$ conversion among the three structures has been explained by its reversible restructuring to form ~2 nm metallic Cu nanoclusters, which are identified as the active sites for the electrocatalysis. Our findings suggest the possibility of controlling catalytic active sites through

molecular structure design, providing insights into strategies for developing high-performance electrocatalyst materials.

## Methods

**Materials.** Nafion perfluorinated resin solution (5 wt% in lower aliphatic alcohols and water), $KHCO_3$ (ACS Reagent 99.7%), and $H_3PO_4$ (ACS Reagent ≥85%) were purchased from Sigma Aldrich. Graphite rod (99.9995%) and Ti foil (0.127 mm, 99.99%) were purchased from Alfa Aesar. CuPc (dye content 99.13%) was purchased from Acros Organics. HCl (ACS Reagent 36.5 ~ 38%) was purchased from J.T. Baker. All materials were used as obtained without further purification. Deionized water (Milli-Q Millipore 18.2 MΩ $cm^{-1}$) was used throughout all the experiments.

**HKUST-1 synthesis.** To synthesize HKUST-1, a solution of 0.252 g of $H_3$btc (1.2 mmol) in 12 ml of water/ethanol (volume ratio = 2:1) was rapidly added into a solution of $Cu(NO_3)_2 \cdot 3H_2O$ (0.145 g, 0.6 mmol) in 12 ml of water under vigorous magnetic stirring (1200 rpm). The mixture was kept under the stirring condition for 120 min at room temperature. The product was collected by repeated ethanol wash and centrifugation for more than five times until the supernatant was colorless. The final product was lyophilized. XRD pattern of HKUST-1 powder is shown in Supplementary Fig. 12.

**[Cu(cyclam)]$Cl_2$ synthesis.** The preparation of [Cu(cyclam)]$Cl_2$ is shown in Supplementary Note 3. The copper salt was used in slightly less amount (0.93 eq relative to cyclam), to ensure the complete consumption of copper ion, thus to ensure the absence of copper ion in the final product. The excess cyclam was removed by recrystallization from isopropanol. To synthesize [Cu(cyclam)]$Cl_2$, A solution of 1,4,8,11-tetraazacyclotetradecane (cyclam, 150 mg, 0.75 mmol) in ethanol (20 ml) was added $CuCl_2 \cdot 2H_2O$ (121 mg, 0.70 mmol, 0.93 eq) in one portion. The color changed immediately from colorless to purple. The resulting solution was stirred at 80 °C under nitrogen atmosphere. After 5 h, the solution was cooled to room temperature and then filtered through a filter paper to remove the insoluble material, whereupon ethanol was removed by rotary evaporation. Isopropanol (~ 40 ml) was then added to the purple residue and the mixture was heated to reflux to completely dissolve the solid. The resulting purple solution was placed in a freezer (− 20 °C) overnight, and the precipitated solid was collected by centrifugation and dried under high vacuum to afford [Cu(cyclam)]$Cl_2$ (228 mg, 90% yield) as a purple powder. The UV–Vis absorption spectrum of [Cu(cyclam)]$Cl_2$ in $CH_3OH$ is shown in Supplementary Fig. 13. High-resolution MS (electrospray) $m/z$ $(M − 2Cl)^{2+}$ calcd for $C_{12}H_{28}CuCl_2N_4$ 146.58, found 146.56; $\lambda_{max}(CH_3OH)$ 260, 528 nm.

**Characterizations.** SEM measurements were performed with a Hitach SU8230 cold field emission SEM microscope. XRD patterns were collected with a Rigaku SmartLab X-ray Diffractometer equipped with a Cu-target X-ray tube ($\lambda = 0.154$ nm) and operated at 40 mA and 44 kV. Absorption spectra were recorded on a Varian Cary 50 Bio UV-visible spectrophotometer. The mass spectral data were obtained from a Thermo Scientific LTQ Orbitrap ELITE mass spectrometer. The sample was directly infused into the mass spectrometry via a syringe pump at 5 μl $min^{-1}$.

**Electrochemical measurements.** Electrochemical experiments were performed on a Bio-Logic VMP3 Multi Potentiostat using a home-made gas-tight two-compartment electrochemical cell. Two milligrams of catalyst materials and 2 mg of mildly oxidized multi-wall CNTs were mixed with 12 μl of 5 wt% Nafion solution and 2 ml of methanol by sonication for more than 30 min to form homogeneous inks. The CNTs were prepared following a modified Hummers method as described in our previous work[20]. For each material, 7.5 μl of the ink was dropped onto a well-polished glassy carbon disk electrode (diameter: 4 mm) and allowed to dry. The catalyst mass loading was 60 μg $cm^{-2}$. A graphite rod and a Ag/AgCl electrode were used as the counter and reference electrodes. The working electrode compartment and the counter electrode compartment were separated by an anion exchange membrane (Selemion DSV). Each compartment contained 12 ml of electrolyte and ~ 18 ml of gas headspace. Pre-purified 0.5 M $KHCO_3$ aqueous solution was used as the electrolyte for all experiments. The electrolyte was purified following a method described in our previous work[15]. Before measurement, the electrolyte was pre-saturated with $CO_2$ by bubbling the gas for 15 min. During measurement, $CO_2$ was continuously bubbled into the electrolyte at a flow rate of 10 s.c.c.m. Current densities were normalized to the geometric area of the glassy carbon electrode. All potentials were referred to the RHE and were recorded with iR compensation.

**Product quantification.** Gas products of electrocatalysis were analyzed by a GC (SRI Multiple Gas Analyzer #5) equipped with molecular sieve 5A and HayeSep D columns with $N_2$ as the carrier gas. Hydrogen was analyzed by a thermal conductivity detector, and carbon monoxide, methane, and ethylene were determined using a flame ionization detector. The peak areas were converted to gas volumes using calibration curves. Liquid products were quantified after electrocatalysis by

[1]H NMR (V600a Varian VNMRS 600 MHz NMR). Electrolyte (700 μl) was mixed with 35 μl of 10 mM dimethyl sulfoxide and 50 mM phenol as internal standards in $D_2O$ for the [1]H NMR analysis.

**In-situ and operando XAS measurements.** In-situ XANES and EXAFS experiments were carried out at beamline 5BM-D of DND-CAT, Advanced Photon Source, Argonne National Laboratory. The working electrodes were prepared by depositing catalysts on ~ 100 μm-thick carbon fiber paper. For HKUST-1 and CuPc, 6.4 mg of material were mixed with 48 μl of 5 wt% Nafion solution and 2 ml of methanol by sonication for more than 30 min to form a homogeneous ink, and then 140 μl of the ink was drop-dried onto a 2.5 × 1.5 $cm^2$ carbon fiber paper (Toray030–30%PTFE) to form a 0.5 × 1 $cm^2$ active area (corresponding to a catalyst mass loading of 1.8 mg $cm^{-2}$). For [Cu(cyclam)]$Cl_2$, 140 μl of a CNT ink (1.6 mg of CNTs mixed with 6.4 μl of 5 wt% Nafion solution and 2 ml of methanol by sonication for more than 30 min) was drop-dried onto a 2.5 × 1.5 $cm^2$ carbon fiber paper (GDS1120) to form a 0.5 × 1 $cm^2$ area, and then 140 μl of 3.2 mg $ml^{-1}$ methanol solution of [Cu(cyclam)]$Cl_2$ was drop-dried onto the CNT area (corresponding to a catalyst mass loading of 1.8 mg $cm^{-2}$). The catalyst electrode was mounted onto a custom-designed in-situ XAS fluorescence cell (Supplementary Fig. 3), as described in our previous study[5,35]. The cell which can contain up to 30 mL of electrolyte was set in a three-electrode configuration. A graphite rod and a Ag/AgCl electrode were used as the counter and reference electrodes, respectively. The same electrolyte was used as described in the Electrochemical measurements session. During the in-situ and operando XAS measurements, $CO_2$ was constantly bubbled at a flow rate of 30 s.c.c.m. All data were collected in a fluorescence mode under various applied potentials controlled by a Gamry Reference-600 electrochemical workstation. A Vortex ME4 detector was used to collect the Cu K fluorescence signal while a Si(111) monochromator scanned the incident X-ray photon energy through the Cu K absorption edge. The monochromator was detuned to 65% of the maximum intensity at the Cu K edge to minimize the presence of higher harmonics. Each selected potential (iR compensated) was held until enough data statistics of XAS were achieved. The X-ray beam was calibrated using a Cu metal foil. Data reduction, data analysis, and EXAFS fitting were performed with the Athena, Artemis, and IFEFFIT software packages. Standard procedures were used to extract the EXAFS data from the measured absorption spectra. The pre-edge background was linearly fitted and subtracted. The post-edge background was determined using a cubic-spline-fit procedure and then subtracted. Normalization was performed by dividing the data by the height of the absorption edge at 50 eV. For quantitative analysis, phase shifts and back-scattering amplitudes were generated by the FEFF calculations based on crystal structures of Cu and CuPc, and were then calibrated through performing the FEFFIT of the EXAFS data of the reference samples, mainly to obtain the amplitude reduction factor ($S_0^2$) values. With $S_0^2$ known, the EXAFS data of the catalyst materials were fitted with such generated phase shifts and amplitudes. Accuracies of the obtained results presented here are as follows: $\Delta N$ (± 10%), $\Delta R$ (± 1%), $\Delta \sigma^2$ (± 10%), and $\Delta E_0$ (± 10%)[34,53,54].

**EXAFS modeling and analysis.** The EXAFS data of Cu foil and CuPc powder were fitted (Supplementary Fig. 7) and the obtained $S_0^2$ values were used as references to calculate the CNs in the analysis of the in-situ EXAFS data. As the in-situ XANES and EXAFS spectra of CuPc show coexistence of Cu(0) and Cu(II) at several applied potentials, it is natural to use the scattering paths from Cu metal and CuPc crystal structures to fit the in-situ EXAFS spectra. To reduce the number of fitting parameters and to increase the information content, co-refinement of a total of 9 EXAFS data sets was performed. Many parameters such as mean-square disorder ($\sigma^2$), energy shift ($E_0$), and scattering path length change ($\Delta R$ or $\alpha$ in $\alpha \star R$) are shared across all the data sets, and only the CNs for each scattering path in each data set are independent and separated. This results in 50 fitting parameters for a total of 175 independent variables. For each data set, there is an average of less than 6 fitting parameters, much less than what is used in conventional EXAFS fitting. All of these fitting parameters were used without any particular constraints. Fitting was done through three shells by taking into account multiple-scattering paths but only the major single scattering paths, namely Cu-$N_1$, Cu-$C_1$, and Cu-$N_2$ for CuPc, as well as Cu-$Cu_1$, Cu-$Cu_2$, and Cu-$Cu_3$ for Cu, illustrated in Supplementary Fig. 8, are listed in Supplementary Table 1.

**Linear combination fit.** Owing to the ensemble average nature of XAS measurements, it is possible to perform a linear combination fit[34] using the reference spectra of Cu foil and CuPc powder to obtain the percentages of Cu nanoclusters and remaining CuPc in the catalyst material. A fit of the XANES spectrum at − 1.06 V overestimates the content of CuPc as the characteristic Cu(II) peak of the fitted spectrum is much higher in intensity than that of the measured spectrum (Supplementary Fig. 10A). A fit of the EXAFS spectrum (Supplementary Fig. 10B) gives 13% of CuPc and 87% of metallic Cu. This might have slightly overestimated the metallic Cu content as in our case there are Cu nanoclusters which have smaller average CNs compared with bulk Cu metal. Furthermore, the CNs of bulk materials in a mixture can reflect their concentrations[42]. The EXAFS analysis show that at − 1.06 V the CN of Cu-$N_1$ is 1.2 ± 0.5, which suggests that the CuPc content is 30% ± 12.5% as the theoretical first shell CN of CuPc is 4. Having taken all the

results into consideration, we estimate that the CuPc catalyst material at −1.06 V contains approximately 20% of CuPc and 80% of metallic Cu by atomic ratio.

**Estimation of Cu nanocluster size at −1.06 V.** Nearest-neighbor CNs of nanoclusters are dependent on cluster sizes. The CNs are nonlinear functions of the cluster diameter if the latter is smaller than 3–5 nm[39]. This property is widely used in EXAFS analyses to determine nanocluster size[39,40,55,56]. Using the Cu cuboctahedral model (Supplementary Fig. 9A) and the strategy reported before[39], we calculated the size-dependent CNs of several scattering paths, as shown in Supplementary Fig. 9B and also listed in Supplementary Table 2. Considering the compositional effect, we scaled the CNs obtained in our EXAFS fits to estimate the nanocluster size[42]. For example, at −1.06 V, the CNs of Cu-Cu$_1$, Cu-Cu$_2$, and Cu-Cu$_3$ are 6.5, 1.4, and 15.8, respectively. Considering that there are 80% of Cu nanoclusters in the catalyst material, the true CNs of the Cu nanoclusters should be 8.1, 1.8 and 19.8, respectively. By checking Supplementary Table 2, these CN values correspond to roughly 1, 0.5, and 4 nm. Considering the errors associated with the obtained CN values, we estimate that the average size of the Cu nanoclusters formed at −1.06 V is 2 ± 1 nm.

**TOF calculation.** To calculate the TOF for the CuPc catalyst, the number of surface sites was estimated based on the size and geometry of the metallic Cu clusters using the equation below:

$$\mu = MN = M\frac{\alpha m N_A}{M_{CuPc}} \tag{1}$$

Where $\mu$ denotes the number of surface sites, $M$ denotes the percentage of surface Cu atoms in a Cu cluster, $N$ denotes the total number of Cu atoms in all the Cu clusters on the electrode, $\alpha$ denotes the percentage of CuPc molecules that have restructured to Cu clusters, $m$ denotes the original mass loading of CuPc (60 μg cm$^{-2}$), $N_A$ denotes the Avogadro constant (6.022 × 10$^{23}$), and $M_{CuPc}$ denotes the molecular mass of CuPc (576.07 g mol$^{-1}$). Here, $\alpha$=80% based on the XAS results. Consider that the Cu clusters are 2 nm cuboctahedra containing 162 surface Cu atoms and a total of 309 Cu atoms, $M$ = 0.524. Consequently, $\mu$ = 2.63 × 10$^{16}$ sites per cm$^2$. TOF was calculated using the equation below:

$$TOF = \frac{j}{ne\mu} \tag{2}$$

Where $j$ is the partial current density for CH$_4$ formation, $n$ is the number of electrons needed to reduce one CO$_2$ molecule to CH$_4$, and $e$ is the elementary charge. $j$, $n$, and $e$ are 13 mA cm$^{-2}$, 8, and 1.602 × 10$^{-19}$ C, respectively. Therefore, the TOF of CH$_4$ for the CuPc catalyst at −1.06 V vs RHE is 0.39 molecules site$^{-1}$ s$^{-1}$.

**DFT calculations on the thermodynamics of Cu-complex demetallation and re-metallation.** DFT geometry optimization calculations were performed using the hybrid functional B3LYP, which includes Becke's three parameter exchange[57] and the Lee, Yang and Parr correlation[58] as implemented in Gaussian 09 (Rev. D.01)[59]. For optimizations and thermochemistry ($T$ = 298.15 K), we used the basis set 6–31 G(d)[60] for all atoms. The optimized molecular structures are shown in Supplementary Fig. 14. For single point calculations, we used the basis set def2TZVP[60] for Cu atoms and 6–311 + G(2df,p) for all other atoms[61]. Solvent correction was implemented using the SMD model with water as the solvent[62]. A hypothetical thermodynamic pathway (Supplementary Note 1) under standard conditions is constructed for reductive demetallation which consists of: (1) dissociation and protonation of the ligand and (2) reduction of the hexaaqua Cu(II) ion into bulk Cu metal. Through the Nernst equation, the standard reduction potential of demetallation can in turn be calculated as $E^0_{CuL/Cu} = E^0_{Cu2+/0} − \Delta G^0/4F$, where L denotes the ligand and $\Delta G^0$ refers to the free energy change of step (1) in the pathway and $E^0_{Cu2+/0}$ refers to the standard reduction potential of Cu$^{2+}$ to Cu(0)[63]. Supplementary Table 3 lists the $\Delta G^0$ values and $E^0_{CuL/Cu}$ for all of the three Cu-complex structures investigated in the study. Re-metallation of the Pc$^{2-}$ ligand is considered to be an oxidation of Cu nanoclusters by H$_2$Pc, as illustrated by the hypothetical thermodynamic pathway shown in Supplementary Note 2. The change of free energy ($\Delta G$) for re-metallation is dependent on the size of Cu nanoparticles and is 7.3 kcal mol$^{-1}$ for bulk Cu metal. The size-dependent cohesive energy of Cu nanoparticles is calculated through the following established equation:[64]

$$\frac{E_c(D)}{E_{cb}} = \exp\left(-\frac{2S_{cb}}{3R}\frac{1}{\frac{D}{D_0}-1}\right) \times \left(1 - \frac{1}{\frac{D}{D_0}-1}\right) \tag{3}$$

where $E_c(D)$ denotes the cohesive energy of a nanoparticle with diameter $D$; $E_{cb}$ denotes the cohesive energy of the bulk crystal; $D_0$ denotes the atomic radius; $R$ is the ideal gas constant; $S_{cb}$ is defined as $E_{cb}/T_{cb}$ where $T_{cb}$ refers to the boiling point of bulk Cu metal. $D_0$ = 0.128 nm, $E_{cb}$ = 336 kJ mol$^{-1}$ and $T_{cb}$ = 2840 K are taken from the literature[65,66]. The critical diameter of Cu nanoparticles is defined as the size at which $\Delta G$ of re-metallation is zero. The critical diameter in the case of CuPc is calculated to be 14.2 nm.

**Data availability**. The data that support the findings of this study are available within the paper and its Supplementary Information file or are available from the corresponding authors upon reasonable request.

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

## Acknowledgements

The work is supported by the National Science Foundation (Grant CHE-1651717), the Doctoral New Investigator grant from the ACS Petroleum Research Fund, the Global Innovation Initiative from Institute of International Education, and the Callahan Faculty Scholar Endowment Fund from Oregon State University. The synthetic work was supported by the U.S. Department of Energy, Chemical Sciences, Geosciences, and Biosciences Division, Office of Basic Energy Sciences, Office of Science (Grant DEFG02-07ER15909). Additional support was provided by a generous donation from the TomKat Foundation. V.S.B. acknowledges support from the Air Force Office of Scientific Research grant FA9550-13-1-0020 and supercomputing time from the NERSC. Y.L. acknowledges financial support from Shenzhen Fundamental Research Funding (JCYJ20160608140827794) and Peacock Plan (KQTD20140630160825828). XAS measurements were done at 5-BM-D of DND-CAT at Advanced Photon Source (APS) of Argonne National Laboratory (ANL). DND-CAT is supported through E. I. duPont de Nemours & Co., Northwestern University, and The Dow Chemical Company. The use of APS of ANL is supported by DOE under Contract Number DE-AC02-06CH11357.

## Author contributions

Z.W., Y.L., Z.F., and H.W. conceived the project. J.J. and X.-F.W. synthesized the materials. Z.W. and S.H. carried out the electrochemical measurements and material characterization. Z.W., M.W., Q.M., and Z.F. carried out the in-situ and operando XAS measurements. Y.W. and K.Y. performed cluster size analysis and DFT calculations. M.W. and Z.F. analyzed the XAS data. Z.W., Y.W., Z.F., and H. W. wrote the manuscript. G.W.B, V.S.B, Y.L., Z.F., and H.W. supervised the project. All authors discussed the results and commented on the manuscript.

## Additional information

**Competing interests:** The authors declare no competing financial interests.

