## [Peer Review File · Nature Communications]

Reviewers' comments:

Reviewer #1 (Remarks to the Author):

The manuscript considers the restructuring of three molecularly structured Cu catalysts with in situ and in operando XAS. Their Cu(II) phthalocyanine exhibited Faradaic efficiency for methane to 60% at -1.06 V vs. RHE with partial current density of 13mA/cm². The improved methane production performance is attributed to the stabilization of Cu active sites with the organic complex. The theoretical calculation strongly support the experimental conclusions, however, the reaction products and novelty may not justify publication in Nature Communications.

(1) The study suggests Cu nanoclusters generated from CuPc under reduction potential improve the selectivity to methanation in CO₂ reduction. The enhance methanation with Cu nanoparticle has been studied previously: e.g. K. Manthiram, B. J. Beberwyck and A. P. Alivisatos, *J. Am. Chem. Soc.*, 136, 13319 (2014). The Faradaic efficiency was reported to be as high as 85%.

(2) Previous studies including work by D. R. Alfonso, D. Kauffman and C. Matranga (*Journal of Chemical Physics*, 144, 184705 (2016)) provides DFT simulations to show the demetallation of metal-ligand complex to create active sites. Other works (e.g. Y. Fang and J. Flake (*J. Am. Chem. Soc.*, 2017, 139 (9), (2017)) also suggest ligands remain near the catalyst surface and play a role in the reduction.

(3) The x axis scale in Figure 2 (a) is different from other XANES spectra and Figure 2 (G)(H)(I) didn't provide the XAS at 0.64V.

(4) In the caption of figure 3, does the author intend to say " before and after the electrolysis"?

Overall the work is high quality and should be published, the only concern is the novelty and significance of these results relative to other works.

Reviewer #2 (Remarks to the Author):

The manuscript submitted by Weng, et al. describes the preparation of catalyst precursors and their corresponding catalytic activity, specifically for the conversion of carbon dioxide to methane. The manuscript is well assembled and the work appears to be well performed. The characterization and reactivity of the three different copper complexes are of interest to the community, especially given the remarkable difference in catalytic activity for the three complexes. I recommend publication in Nature Communications, pending correction of a few relatively minor issues.

1. The take-home message could be better conveyed in the abstract. One route towards that goal would be to state earlier the catalytic activity, and then follow the activity with the explanations thereof (currently earlier in the abstract).
2. Some of the figures, such as Figure 3 should be better labeled, such as including some indication in the graphic itself of which complex is being characterized, as well as indicating which images are before and which are after, rather than relying solely on the caption.
3. On Page 12, line 238, “is a major contributor to” should be correct to something like “appears to be a major contributor to”
4. The verb tense used in the abstract should be corrected, as it is currently written in the present/future tense to description observations or outcomes that happened in the past.

Reviewer #3 (Remarks to the Author):

The present manuscript focus on the restructuring of Cu complexes in CO₂ electroreduction processes. The study is developed by in-situ and operando X-ray absorption spectroscopy. The results showed that the use of Cu(II)phthalocyanine complex is advantageous in comparison to the other structures tested. The activity of this material for the CO₂-to-CH₄ reaction is explained by a reversible restructuring to 2 nm metallic nanoclusters that act as active sites.

Technically speaking, this study is valuable and the results will stimulate the researchers in the field. I also think the report is written clearly and in a highly comprehensive manner. I also found the report an important contribution to the field, providing new insights for the development of highly active electrocatalytic materials. Overall, I believe it may be suitable for publication in Nature Communications after minor revision. These are the comments from my side:

- The reaction for CH₄ is challenging due to the large variety of products that could be generated at high overpotentials and due to the parasitic hydrogen evolution reaction which lowers the overall yield. In this regard, I wonder if in the liquid phase analyses other reaction species (methanol, ethanol...) were detected. This data would be of help to hypothesize the reaction mechanisms.
- The rate for CH₄ formation are not shown. These values may give valuable information to evaluate the performance of the system. Could the authors present CH₄ yield normalized by catalyst loading, available reaction area and charge passed through the system? Could the authors show the experimental error in the obtained results?

Responses to Reviewers

Replies to Reviewer #1 and revisions made:

The manuscript considers the restructuring of three molecularly structured Cu catalysts with in situ and in operando XAS. Their Cu(II) phthalocyanine exhibited Faradaic efficiency for methane to 60% at -1.06 V vs. RHE with partial current density of 13 mA/cm². The improved methane production performance is attributed to the stabilization of Cu active sites with the organic complex. The theoretical calculation strongly support the experimental conclusions, however, the reaction products and novelty may not justify publication in Nature Communications. Overall the work is high quality and should be published, the only concern is the novelty and significance of these results relative to other works.

We thank the reviewer for the positive comments on this work. After reading the papers raised by the reviewer and comparing them carefully with this work, we confirm that this work fundamentally differs from what has been reported before. Please see below our point-by-point responses to the reviewer's concerns.

(1) The study suggests Cu nanoclusters generated from CuPc under reduction potential improve the selectivity to methanation in CO₂ reduction. The enhance methanation with Cu nanoparticle has been studied previously: e.g. K. Manthiram, B. J. Beberwyck and A. P. Alivisatos, J. Am. Chem. Soc., 136, 13319 (2014). The Faradaic efficiency was reported to be as high as 85%.

Response:

We thank the reviewer for raising the reference (J. Am. Chem. Soc., 2014, 136, 13319), which was cited as ref 44 in our original manuscript.

The Cu nanoparticle catalyst in the reference shows a FE of 46% and a partial current density of 1.2 mA/cm² for CH₄ production at -1.05 V vs RHE (please see the figure below adapted from the reference), which is obviously worse than our CuPc catalyst (FE 66% and partial current density 13 mA/cm²) at almost identical potential (-1.06 V vs RHE). The highest FE for the Cu nanoparticle catalyst, 77%, is achieved at a much larger overpotential (-1.35 V vs RHE), and even at such a negative potential, the current density is less than 10 mA/cm². Therefore, our catalyst is at least arguably more active, and clearly represents one of the most efficient catalysts for electrochemical reduction of CO₂ to CH₄.

The high activity of our catalyst is just one significant aspect of this work. We would like to emphasize that the most noteworthy knowledge obtained in this work is the discovery of the reversible restructuring of CuPc molecules to ~2 nm metallic Cu clusters under working conditions, which explains the high activity of Cu-tetrapyrrole

materials for electrochemical CO₂ reduction to CH₄. Such a restructuring behavior has never been reported before for any metal-complex-based CO₂ reduction electrocatalyst.

Figure 2. Comparison of current densities and Faradaic efficiencies for n-Cu/C and copper foil electrodes. (A) Total current density, demonstrating that n-Cu/C has greater overall reduction activity than the copper foil. (B) Faradaic efficiency for methane, in which it is evident that n-Cu/C is more selective for methane than the copper foil. (C) Methanation current density, in which the combined effect of the improved current density and Faradaic efficiency on n-Cu/C is apparent. (D) Faradaic efficiency for hydrogen as a function of potential, showing suppressed hydrogen evolution on the n-Cu/C catalyst.

(2) Previous studies including work by D. R. Alfonso, D. Kauffman and C. Matranga (Journal of Chemical Physics, 144, 184705 (2016)) provides DFT simulations to show the demetallation of metal-ligand complex to create active sites. Other works (e.g. Y. Fang and J. Flake (J. Am. Chem. Soc., 2017, 139 (9), (2017)) also suggest ligands remain near the catalyst surface and play a role in the reduction.

Response:

We thank the reviewer for raising the two references to our attention. While both works unravel an important role played by surface ligands in electrocatalytic CO₂ reduction, they fundamentally differ from the key message we deliver in this manuscript, as detailed in the following three paragraphs.

In this work, we find that CuPc reversibly restructures to ~2 nm metallic Cu clusters under working conditions, and the clusters, which are detected by *in-situ* and *operando* XAS measurements, are the real active species responsible for the observed high catalytic activity of the CuPc material for electrochemical CO₂ reduction to CH₄. Our material is a metal-complex molecular solid, and the reaction is CO₂ conversion

to CH₄.

The first reference raised by the reviewer (Journal of Chemical Physics, 2016, 144, 184705) is a computational work which suggests that the non-capped surface atoms of Au clusters are active for CO₂ electroreduction. The catalyst material model is Au clusters with capping ligands, and the reaction is CO₂ conversion to CO.

The second reference raised by the reviewer (J. Am. Chem. Soc., 2017, 139, 3399) is on the effects of using tethered ligands to modify electrocatalytic properties. The catalyst is a bulk Au electrode modified with thiol molecules, and the reaction is CO₂ conversion to CO or formic acid.

Therefore, these two previous works do not have any considerable overlap with this work and should not undermine the novelty of this work. Having that said, these two references do give useful insights into the roles of ligands in electrocatalytic CO₂ reduction. We have cited them in the revised manuscript and we again thank the reviewer for bringing them to our attention.

(3) The x axis scale in Figure 2 (a) is different from other XANES spectra and Figure 2 (G)(H)(I) didn't provide the XAS at 0.64V.

Response:

We thank the reviewer for catching the different scales in the figure. We have changed the horizontal scale for Figure 2a so that all the XANES graphs now have the same energy scale.

We were not able to record reasonable XAS spectra for [Cu(cyclam)]Cl₂ back to 0.64 V because some of the Cu metal formed at -1.06 V detached from the electrode surface and caused much noise in the XAS spectrum. This information was given in the caption of Figure S4 in the original SI.

(4) In the caption of figure 3, does the author intend to say “before and after the electrolysis”?

Response:

Yes, and we have changed the caption accordingly.

Replies to Reviewer #2 and revisions made:

The manuscript submitted by Weng, et al. describes the preparation of catalyst precursors and their corresponding catalytic activity, specifically for the conversion of carbon dioxide to methane. The manuscript is well assembled and the work appears to be well performed. The characterization and reactivity of the three different copper complexes are of interest to the community, especially given the remarkable difference in catalytic activity for the three complexes. I recommend publication in Nature Communications, pending correction of a few relatively minor issues.

We greatly appreciate the reviewer's positive comments on this work. Please see below our point-by-point responses to the reviewer's concerns.

1. The take-home message could be better conveyed in the abstract. One route towards that goal would be to state earlier the catalytic activity, and then follow the activity with the explanations thereof (currently earlier in the abstract).

Response:

We thank the reviewer for the helpful suggestion. We have made changes to the abstract accordingly, by stating the catalytic reactivity before the restructuring results. The revised abstract is shown below.

“Restructuring-induced catalytic activity is an intriguing phenomenon of fundamental importance to rational design of high-performance catalyst materials. We studied three Cu-complex materials for electrocatalytic CO₂ reduction. Among them, the Cu(II) phthalocyanine (CuPc) exhibits by far the highest activity for reducing CO₂ to methane with a Faradaic efficiency of 66% and a partial current density of 13 mA cm⁻² at the potential of -1.06 V vs RHE. Utilizing *in-situ* and *operando* X-ray absorption spectroscopy measurements to capture the dynamic changes in the oxidation state as well as in the coordination sphere of the Cu center, we found that under the working conditions CuPc reversibly forms ~2 nm metallic Cu clusters which catalyzes the CO₂-to-methane conversion. Density functional calculations rationalize the restructuring behavior and attribute the reversibility to the strong divalent metal ion-ligand coordination in the CuPc molecular structure and the small size of the generated Cu clusters under the reaction conditions.”

2. Some of the figures, such as Figure 3 should be better labeled, such as including some indication in the graphic itself of which complex is being characterized, as well as indicating which images are before and which are after, rather than relying solely on the caption.

Response:

We appreciate the reviewer's suggestion. We have modified Figure 3 accordingly. The new figure is shown below. We have added complex identity information to the graphs in Figure 2 as well.

Figure 3 | XRD and SEM characterizations of the three catalyst materials before and after electrocatalysis. XRD patterns of (A) HKUST-1, (B) [Cu(cyclam)]Cl₂ and (C) CuPc before and after electrocatalysis. The background diffraction patterns are from carbon paper substrates. SEM images of the (D, G) HKUST-1, (E, H) [Cu(cyclam)]Cl₂ and (F, I) CuPc catalyst materials (D, E, F) before and (G, H, I) after electrocatalysis. Scale bars: (D, E, G, H) 200 nm; (F, I) 100 nm.

3. On Page 12, line 238, “is a major contributor to” should be correct to something like “appears to be a major contributor to”

Response:

We have made the change as the reviewer suggested.

4. The verb tense used in the abstract should be corrected, as it is currently written in the present/future tense to description observations or outcomes that happened in the

past.

Response:

Following the reviewer's suggestion, we have corrected the verb tense in the abstract.

Replies to Reviewer #3 and revisions made:

The present manuscript focus on the restructuring of Cu complexes in CO₂ electroreduction processes. The study is developed by in-situ and operando X-ray absorption spectroscopy. The results showed that the use of Cu(II)phthalocyanine complex is advantageous in comparison to the other structures tested. The activity of this material for the CO₂-to-CH₄ reaction is explained by a reversible restructuring to 2 nm metallic nanoclusters that act as active sites.

Technically speaking, this study is valuable and the results will stimulate the researchers in the field. I also think the report is written clearly and in a highly comprehensive manner. I also found the report an important contribution to the field, providing new insights for the development of highly active electrocatalytic materials. Overall, I believe it may be suitable for publication in Nature Communications after minor revision. These are the comments from my side:

We greatly appreciate the reviewer's positive comments on this work. Please see below our point-by-point responses to the reviewer's concerns.

(1) The reaction for CH₄ is challenging due to the large variety of products that could be generated at high overpotentials and due to the parasitic hydrogen evolution reaction which lowers the overall yield. In this regard, I wonder if in the liquid phase analyses other reaction species (methanol, ethanol...) were detected. This data would be of help to hypothesize the reaction mechanisms.

Response:

We understand the reviewer's concern. As shown in Fig. 1 D and 1E, formic acid is the only liquid product we detected.

(2) The rate for CH₄ formation are not shown. These values may give valuable information to evaluate the performance of the system. Could the authors present CH₄ yield normalized by catalyst loading, available reaction area and charge passed through the system? Could the authors show the experimental error in the obtained results?

Response:

We appreciate the reviewer's helpful suggestions.

In the original manuscript, the rate of CH₄ formation was expressed as the partial current density, which as the reviewer pointed out is not straightforward. Following the reviewer's suggestion, we calculated the CH₄ production yield normalized to the catalyst mass loading ($0.36 \text{ mmol}\cdot\text{s}^{-1}\cdot\text{g}_{\text{CuPc}}^{-1}$ and $3.2 \text{ mmol}\cdot\text{s}^{-1}\cdot\text{g}_{\text{Cu}}^{-1}$), the active surface area ($0.39 \text{ molecules}\cdot\text{s}^{-1}\cdot\text{site}^{-1}$) and the total charge ($0.86 \mu\text{mol}\cdot\text{C}^{-1}$).

To the manuscript we have added “At –1.06 V, a maximum Faradaic efficiency of 66% together with a partial current density of 13 mA cm⁻² is achieved for CO₂ conversion to CH₄ (Fig. 1D, 1E), corresponding to a CH₄ formation rate of 0.36 mmol·s⁻¹·g_{CuPc}⁻¹ and 0.86 μmol·C⁻¹.”, and “The corresponding CH₄ production rate and turnover frequency (TOF) are 3.2 mmol·s⁻¹·g_{Cu}⁻¹ and 0.39 molecules·s⁻¹·site⁻¹, respectively.”.

To the SI, we have added the following details for TOF calculation: “**TOF Calculation.** To calculate the TOF for the CuPc catalyst, the number of surface sites was estimated based on the size and geometry of the metallic Cu clusters using the equation below:

$$\mu = MN = M \frac{\alpha m N_A}{M_{CuPc}}$$

Where μ denotes the number of surface sites, M denotes the percentage of surface Cu atoms in a Cu cluster, N denotes the total number of Cu atoms in all the Cu clusters on the electrode, α denotes the percentage of CuPc molecules that have restructured to Cu clusters, m denotes the original mass loading of CuPc (60 μg cm⁻²), N_A denotes the Avogadro constant (6.022×10²³) and M_{CuPc} denotes the molecular mass of CuPc (576.07 g mol⁻¹). α equals to 80% based on the XAS results. Consider that the Cu clusters are 2 nm cuboctahedra containing 162 surface Cu atoms and a total of 309 Cu atoms, M equals to 0.524. Consequently, μ is equal to 2.63×10¹⁶ sites cm⁻². TOF was calculated using the equation below:

$$TOF = \frac{j}{ne\mu}$$

Where j is the partial current density for CH₄ formation, n is the number of electrons needed to reduce one CO₂ molecule to CH₄, and e is the elementary charge. j , n and e are 13 mA cm⁻², 8 and 1.602×10⁻¹⁹ C, respectively. Therefore, the TOF of CH₄ for the CuPc catalyst at –1.06 V vs RHE is 0.39 molecules·site⁻¹·s⁻¹.”

Following the reviewer’s suggestion, we have added error bars to all the electrocatalytic results. The new figures are shown below.

Figure 1 | Molecular Structures of three Cu-complex materials and their electrocatalytic performance for CO₂ reduction. Molecular structures of (A) CuPc, (B) HKUST-1 and (C) [Cu(cyclam)]Cl₂. Potential-dependent (D) Faradaic efficiencies and (E) partial current densities of products for CO₂ electroreduction reaction catalyzed by CuPc. Comparison of (F) Faradaic efficiency and (G) partial current density distributions among CO₂ electroreduction reactions catalyzed by the three materials at -1.06 V vs RHE. Error bars represent the standard deviations from multiple measurements.

Supplementary Figure 1. Potential-dependent (A) Faradaic efficiencies and (B) partial current densities of gas products for CO₂ electroreduction catalyzed by HKUST-1.

Supplementary Figure 2. Potential-dependent (A) Faradaic efficiencies and (B) partial current densities of gas products for CO₂ electroreduction catalyzed by [Cu(cyclam)]Cl₂.

REVIEWERS' COMMENTS:

Reviewer #1 (Remarks to the Author):

The rebuttal to the original review is well considered and presents a reasonable answer to this reviewer's concerns. Overall this is a very good paper. My central remaining concerns are novelty and significance. For example The work by Royce Murray et al in Langmuir, 2000, 16 (16), pp 6682–6688 showed reversible formation of Cu aggregates from complexes. Other related works also show similar rates of methane formation at nanoparticle Cu electrodes. The combination here is novel, but the the results are not necessarily groundbreaking (~70% FE the CH₄ and ~13 mA/cm²). The results and analyses are clearly insightful, but I struggle to judge the significance of this work relative to other works in Nature Communications. This is probably a greater concern to the editor rather than a reviewer.

Reviewer #2 (Remarks to the Author):

The authors have suitably revised the manuscript to address the concerns I had previously raised. In my opinion, the manuscript is suitable for publication in Nature Communications.

Reviewer #3 (Remarks to the Author):

I believe all comments aroused by the reviewer have been tackled correctly and the manuscript is ready for publication.

Response to Reviewer #1's comments

The rebuttal to the original review is well considered and presents a reasonable answer to this reviewer's concerns. Overall this is a very good paper. My central remaining concerns are novelty and significance. For example The work by Royce Murray et al in Langmuir, 2000, 16 (16), pp 6682–6688 showed reversible formation of Cu aggregates from complexes. Other related works also show similar rates of methane formation at nanoparticle Cu electrodes. The combination here is novel, but the the results are not necessarily groundbreaking (~70% FE the CH₄ and ~13 mA/cm²). The results and analyses are clearly insightful, but I struggle to judge the significance of this work relative to other works in Nature Communications. This is probably a greater concern to the editor rather than a reviewer.

Reply:

This time the reviewer raised a new reference to support his/her claim against the novelty of our work. Unfortunately, the reference (Langmuir, 2000, 16 (16), pp 6682–6688) is not even relevant here. The research reported in the reference is all about assembly/aggregation of Au nanoclusters using Cu²⁺ as linkers. It has nothing to do with reduction/demetallation of Cu-containing complexes to form metallic Cu nanoclusters. Nor does it relate to restructuring under electrochemical conditions. It is not connected with electrocatalytic CO₂ reduction reactions either. With due respect to the reviewer, the authors doubt if the reviewer had read this particular reference carefully, if at all, before it was cited in the review comments to attack the novelty and significance of our work.

As we have emphasized several times in the manuscripts and our rebuttal letters, the most noteworthy knowledge obtained in this work is the discovery of the reversible restructuring of CuPc molecules to ~2 nm metallic Cu clusters under working conditions, which explains the high activity of Cu-tetrapyrrole materials for electrochemical CO₂ reduction to CH₄. Such a restructuring behavior has never been reported before for any metal-complex-based CO₂ reduction electrocatalyst.